# Clinical correlates of workplace injury occurrence and recurrence in adults

**Zhaoyi Chen**[1,2], **Mattia Prosperi**[1], **Jiang Bian**[2], **Jae Min**[1], **Mo Wang**[3], **Chang Li**[4]*

**1** Department of Epidemiology, University of Florida, Gainesville, Florida, United States of America,
**2** Department of Health Outcomes and Biomedical Informatics, University of Florida, Gainesville, Florida, United States of America, **3** Department of Management, University of Florida, Gainesville, Florida, United States of America, **4** School of Business Administration, Zhejiang Gongshang University, Hangzhou, China

* lichang@zjgsu.edu.cn

## Abstract

### Objectives

To examine the morbidities associated with workplace injury and to explore how clinical variables modify the risk of injury recurrence.

### Methods

A case-control study was designed using Florida's statewide inpatient, outpatient, and emergency visits data obtained from the Healthcare Cost and Utilization Project. We included adults who were admitted for a workplace injury (WPI) or injury at other places (IOP), and a matched population of random controls without WPI/IOP. The associations between WPI and clinical morbidities were assessed by univariate and multivariable regression, ranking predictors by information gain, area under the receiver operating characteristic (AUROC), and odds ratios. We analyzed WPI recurrence using survival methods (Kaplan-Meier, Cox regression, survival decision trees) and developed prediction models via regularized logistic regression, random forest, and AdTree. Performance was assessed by 10-fold cross-validation comparing AUROC, sensitivity, specificity, and Harrell's c-index.

### Results

A total of 80,712 WPI, 161,424 IOP, and 161,424 control patients were included; 485 distinct clinical diagnostic and 160 procedure codes were analyzed after filtering. Acute bronchitis and bronchiolitis, sprains and strains of shoulder and upper arm, ankle and foot, or other and unspecified parts of back, accidents caused by cutting and piercing instruments or objects, and overexertion and strenuous movements were identified as important consequences of WPI. The prediction models of injury recurrence identified several key factors, such as insurance type and prior injury events, although none of the models exhibited high predictive performance (best AUROC = 0.60, best c-index = 0.62).

### Conclusions

WPI is associated to diverse serious physical comorbidity burden. There are demographic, social and clinical comorbidity components associated to the risk of WPI recurrence,

**Data Availability Statement:** The data used in this study are available for purchase from the Healthcare Cost and Utilization Project (https://www.hcup-us.ahrq.gov/) after completion of requirement trainings and data use agreement.

**Funding:** This work was supported by the grant UL1TR001427 and UL1TR002389 from NIH-NCATS, by the University of Florida (UF) One Health Center, and by UF "Creating the Healthiest Generation" Moonshot initiative, which is supported by the UF Office of the Provost, UF Office of Research, UF Health, UF College of Medicine and UF Clinical and Translational Science Institute. The funders had no role in study design, data collection and analysis, decision to publish, or preparation of the manuscript.

**Competing interests:** The authors have declared that no competing interests exist.

although their predictive value is moderate, which warrants future investigation in other information source domains, e.g. deepening into the environmental and societal sphere.

## Introduction

Workplace injury is a public health concern. According to the United States' Bureau of Labor Statistics, in 2016 in there were approximately 2.9 million nonfatal workplace injuries and illnesses reported by private industry employers; additionally, the number of fatal injuries was more than 5,000 [1]. Although the overall rates of workplace injuries have been steadily decreasing in the U.S. in the past decade, they still pose a substantial economic burden. It is estimated that nonfatal workplace injuries cost nearly $60 billion each year in direct compensation [2,3].

Individual consequences from workplace injuries lead to substantial personal life and public health burden. Previous studies have shown how occupational injuries are associated with increased medical care encounters and health insurance claims. Workplace injuries often entail a variety of psychological and behavioral responses, including stress, reluctance of return to work, and other personal and social afflictions [4].

Recently, the National Institute for Occupational Safety and Health suggested a framework for assessing workplace injury burden, which includes four main approaches: (1) utilizing multiple information source domains; (2) taking a broader view of injuries and related diseases; (3) assessing the impact of the entire working-life continuum; and (4) applying the comprehensive concept of "well-being" [5]. Following this framework, we aimed to examine the short-term and long-term consequences of workplace injury, collating information from the socio-demographic and clinical domains, using a large state-wide healthcare database of the US State of Florida. The overarching goal was to investigate which clinical consequences are associated with workplace injury and to explore how sociodemographic and clinical variables modify the risk of injury recurrence.

## Methods

This study was evaluated by the University of Florida's institutional review board as exempt (protocol no. IRB201701906).

The Healthcare Cost and Utilization Project (HCUP)'s State Inpatient Databases (SID), State Ambulatory Surgery and Services Databases (SASD), and State Emergency Department Databases (SEDD) for the state of Florida, US between 2005 and 2014 were utilized [6]. The SID, SASD and SEDD contain anonymized, longitudinally-linked inpatient, outpatient, and emergency room visits data, including patients' demographics, insurance, diagnoses, and procedures for each hospital visit. Between 2005 and 2014, diagnoses and procedures have been encoded using the International Classification of Diseases version 9 (ICD-9) ontology.

Our study included patients aged 18 years and older with at least three years of medical records prior to baseline and with at least five years of follow-up. The rationale behind exclusion of individuals younger than 18 came from US federal and Florida state laws that regulate the employability of minors and the maximum number of working hours per week. The lengths of prior medical history and of follow up were chosen to assure stability in the areas of residence as well as detailed characterization of health statuses before and after injuries. Workplace injury (WPI) was defined by ICD-9 diagnostic code E846, E849.1, E849.2, and E849.3. There were two comparison groups in our analysis: (1) those who had injuries at other place (IOP)

were defined as any with diagnoses codes E849.0 to E849.9 (except E849.1, E849.2, or E849.3) and (2) a group of patients without any injuries who met the inclusion criteria as random controls. As long as a patient had one WPI, it was assigned to the WPI group; but if any IOP occurred before, it was accounted for. If one subject had more than WPI, the first one recorded in the data base was used to set the baseline date, while the others would count as recurrence (unless they were readmissions, see below). The two comparison groups were extracted in a 2:1 ratio with the WPI sample and were matched on the distribution of diagnostic years.

The observation unit of this study was the individual patient. For each patient, we associated their diagnoses and procedures recorded before the baseline or during the follow-up using three-digit ICD-9 codes. We also calculated the Charlson's comorbidity index (CCI) [7,8] before and after WPI/IOP or the matching date for random controls. Socio-demographic variables included race/ethnicity, insurance status, and area deprivation index [9] associated with the ZIP code of residence at baseline. Diagnostic codes and procedures recorded in less than 2.5% of the WPI group were removed.

To explore the health consequences of WPI, we examined the association between injury status and clinical comorbidities by multivariable logistic regression, with WPI status as independent variable and each of ICD-9 codes as dependent variable after the occurrence of WPI/IOP, i.e. up to 999 for standard ICD-9 codes and 290 for supplemental V&E code. The models were adjusted by age, gender, race, insurance, CCI, and status of the corresponding ICD-9 code prior to the occurrence of WPI/IOP. After fitting all multivariate logistic models, we ranked separately: (a) adjusted odds ratios (OR) of injury status variable, (b) information gain, and (c) area under the receiver operating characteristic (AUROC) of each model. Specifically, the OR measures quantifies the strength of the association (increased occurrence) between two variables. The information gain measures reduction in entropy of one variable by knowing another variable; thus, the less information is lost, the higher the quality of that variable. The AUROC measures the discriminatory power of a variable or model. We then combined these three measurements to create a more robust index to identify importance of health consequences [10].

Finally, we built predictive models for injury recurrence in the WPI group using prior information. WPI recurrence was defined as any WPI diagnosis recorded at least 30 days after the first WPI diagnosis. We decided to use such time window because a WPI diagnosis within the time window could have been a readmission for the same injury. In fact, the Centers for Medicare & Medicaid Services (CMS) define a hospital readmission as "an admission to an acute care hospital within 30 days of discharge from the same or another acute care hospital." Logistic regression with least absolute shrinkage and selection operator (LASSO) regularization, random forest, and AdTree methods were used to predict whether a patient would ever have an injury recurrence (ignoring time-to-event and censoring). Then, survival models (accounting for time-to-event and right censoring) were fit, namely a Cox regression with stepwise selection and a survival tree with log-rank split. Performance was assessed via 10-fold cross-validation, comparing AUROC, sensitivity and specificity, and Harrell's c-index. All statistical analyses were conducted using R and its packages, including glmnet, survival, party, ggplot, survminer, RWeka and pROC [11].

## Results

The 2005–2014 HCUP SID/SASD/SEDD for the State of Florida, after data merging and cleaning, contained a total of 21,091,289 distinct patients. There were 80,712 patients who experienced at least one WPI, 161,424 with IOP, and 161,424 random controls. Among WPI, the vast majority had the ICD code of E849.3 accidents occurring in industrial places and premises

**Table 1. Characteristics of the study population.**

| | Workplace injury | Injury at other places | Random controls |
|---|---|---|---|
| | N (%) | | |
| Total | 80712 | 161424 | 161424 |
| Female | 33925 (42.03%) | 90891 (56.3%) | 91960 (57.0%) |
| Race | | | |
| White | 47108 (58.4%) | 99818 (61.8%) | 105667 (65.5%) |
| Black | 16119 (20.0%) | 28942 (17.9%) | 35623 (22.1%) |
| Hispanic | 14782 (18.3%) | 27890 (17.3%) | 14518 (9.0%) |
| Asian or Pacific Islander | 586 (0.7%) | 927 (0.6%) | 1385 (0.9%) |
| Native American | 135 (0.2%) | 230 (0.1%) | 171 (0.1%) |
| Other | 1982 (2.5%) | 3617 (2.2%) | 4060 (2.5%) |
| Insurance | | | |
| Medicaid | 4595 (5.7%) | 79547 (49.3%) | 70858 (43.9%) |
| Medicare | 6598 (8.2%) | 16073 (10.0%) | 16081 (10.0%) |
| Private | 12653 (15.7%) | 41549 (25.7%) | 48444 (30.0%) |
| Self-pay | **18068 (22.4%)** | 14315 (8.9%) | 12965 (8.0%) |
| No charge | 1099 (1.4%) | 2865 (1.8%) | 4886 (3.0%) |
| Other | **37692 (46.7%)** | 7065 (4.4%) | 7183 (4.5%) |
| Missing | 7 (0.01%) | 21 (0.01%) | 1007 (0.6%) |
| Most frequent residence area (zipcode) | Broward County (33311) (1.0%) | Jacksonville (33209) (1.0%) | Jacksonville (32209) (6.1%) |
| | median (IQR) | | |
| Age | 37 (26, 47) | 56 (39, 71) | 57 (41, 68) |
| Years of prior medical history available | 4 (4, 7) | 5 (4, 7) | 6 (4, 7) |
| Charlson's comorbidity index | 0 (0, 1) | 1 (0, 3) | 0 (0, 1) |
| Area deprivation index | 106.3 (101.1, 109.8) | 105.1 (98.8, 109.2) | 104.6 (99.2, 108.4) |
| Year of diagnosis | 2012 (2010, 2013) | 2013 (2011, 2014) | 2012 (2010, 2013) |

(79,564, 98.58%), while the frequency of mine and quarry accidents (E849.2) farm accidents, (E849.1), and accidents involving powered vehicles used solely within the buildings and premises of industrial or commercial establishment (E846), which are not explicitly WPI–but could be related–were 327 (0.41%), 729 (0.90%), and 92 (0.11%), respectively. Among the IOP, the most common ICD codes were: E849.0 home accidents (84,147, 52.13%), E849.7 Accidents occurring in residential institution (39,491, 24.46%), and E849.5 Street and highway accidents (26.832, 16.62%).

A total of 485 unique three-digit ICD-9 diagnostic codes and 160 ICD-9 procedure codes were identified (all above 2.5% frequency in the WPI group). **Table 1** displays population characteristics stratified by outcome group. There was a higher proportion of males, Black African American and Hispanic ethnicity in the WPI group as compared to IOP group. The proportion of Hispanics was also higher in the WPI group than in the random controls. The patients in the WPI group had a median age of 37 and a median CCI of 0; they were younger and had fewer comorbidities than the IOP group (median age 56 and median CCI 1), while the random controls had a comparable median age of 57 and a median CCI of 0. The WPI group had substantially higher proportion of self-payers (and insurance types other than federal or private) as compared to the other two groups. The patients in the WPI group also resided in areas with a median deprivation index higher than that of the other groups (106.3 in WPI vs. 105.1 in IOP and 104.6 in control).

Among the WPI patients, the most common diagnosis prior to baseline was non-dependent drug abuse (ICD-9 code 305), followed by symptoms involving respiratory system and other

**Table 2. Top-10 most frequent diagnosis in cases (before/after diagnosis).**

|  | workplace injury | other injury | random control |
|---|---|---|---|
| *Top-frequency diagnoses before injury* |  |  |  |
| (401) Essential hypertension | 27.9% | 59.5% | 50.7% |
| (780) General symptoms | 26.6% | 41.1% | 26.9% |
| (272) Disorders of lipoid metabolism | 13.9% | 40.0% | 33.6% |
| (786) Symptoms involving respiratory system and other chest symptoms | 31.5% | 38.0% | 29.3% |
| (276) Disorders of fluid, electrolyte, and acid-base balance | 13.7% | 36.4% | 22.9% |
| (305) Nondependent abuse of drugs | 32.7% | 31.1% | 23.1% |
| (789) Other symptoms involving abdomen and pelvis | 28.3% | 30.5% | 25.7% |
| (530) Diseases of esophagus | 14.8% | 32.2% | 25.8% |
| Most frequent injury code (%) | E849.3 (98.6) | E849.5 (16.6) | n/a |
| *Top-frequency diagnoses after injury* |  |  |  |
| (401) Essential hypertension | 25.1% | 43.9% | 34.2% |
| (V58.6) Long-term (current) drug use | 14.7% | 30.1% | 22.8% |
| (272) Disorders of lipoid metabolism | 12.6% | 29.6% | 22.7% |
| (276) Disorders of fluid, electrolyte, and acid-base balance | 11.2% | 28.9% | 17.1% |
| (780) General symptoms | 17.6% | 27.0% | 16.3% |
| (305) Nondependent abuse of drugs | 23.5% | 19.7% | 13.7% |
| (786) Symptoms involving respiratory system and other chest symptoms | 19.8% | 21.9% | 15.9% |
| (724) Other and unspecified disorders of back | 17.3% | 16.2% | 10.7% |
| (789) Other symptoms involving abdomen and pelvis | 17.3% | 16.7% | 12.9% |
| (530) Diseases of esophagus | 11.9% | 22.8% | 16.7% |

chest symptoms (786), and essential hypertension (401). The most frequent post- WPI diagnoses were essential hypertension (401), non-dependent drug abuse (305), and general symptoms (780), as shown in **Table 2**.

When analyzing deaths, the Kaplan-Meier estimate yielded a four-year survival probability of 85.8% for the WPI group, 93.8% for the IOP group, and 98.2% for the control groups (**Fig 1**).

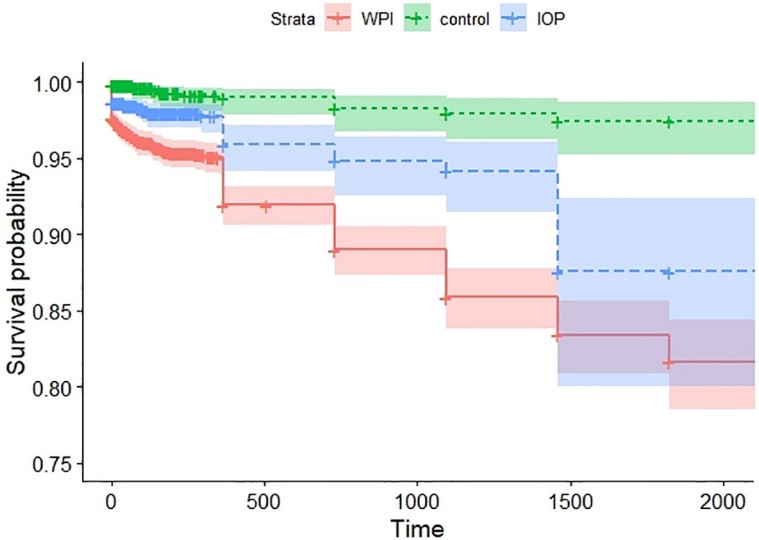

**Fig 1. Survival probability (event = death) in the WPI, IOP and random control groups.**

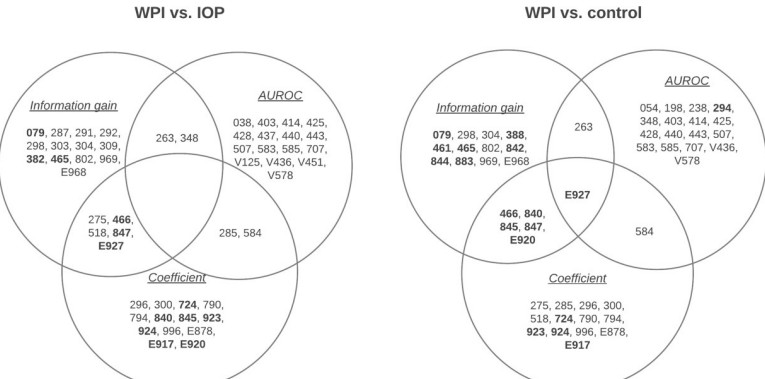

**Fig 2. Venn diagram showing the top-ranked (by AUROC, information gain and odds ratio) ICD-9 codes (i.e. clinical diagnoses) found comparing the WPI vs. IOP group and the WPI vs. random controls.** Variables in boldface represent higher frequency in the WPI group.

Next, we assessed the health consequences of WPI by evaluating the clinical diagnoses made after the injury (or after the corresponding baseline time for the random controls). After Bonferroni adjustment, at significance level of 0.05, a total of 166 variables were identified comparing WPI vs. IOP, and 176 variables for WPI vs. random controls. **Fig 2** shows the top-20 ICD-9 codes identified by merging the three distinct measurements of information gain, AUROC, and odds ratio (as absolute regression coefficient in the linear scale). **Table 3** further

**Table 3. Health consequences differentially associated among WPI, IOP and random controls.** Variable are ranked on the basis of combined AUROC, information gain, and odds ratio (OR). ORs are indicated along with their 95% confidence intervals (CI).

| ICD9 code | condition | vs. injury | | | vs. control | | |
|---|---|---|---|---|---|---|---|
| | | information gain | AUROC | OR (95% CI) | information gain | AUROC | OR (95% CI) |
| 263 | Other and unspecified protein-calorie malnutrition | 85340.41 | 0.640 | 0.985 (0.982, 0.988) | 85340.41 | 0.620 | 0.990 (0.987, 0.993) |
| 275 | Disorders of mineral metabolism | 85231.76 | 0.608 | 0.982 (0.978, 0.987) | | | |
| 285 | Other and unspecified anemias | 85241.44 | 0.633 | 0.971 (0.965, 0.977) | | | |
| 348 | Other conditions of brain | 85236.02 | 0.642 | 0.985 (0.982, 0.989) | | | |
| **466** | **Acute bronchitis and bronchiolitis** | **85216.51** | **0.526** | **1.022 (1.018, 1.027)** | **95416.88** | **0.534** | **1.022 (1.018, 1.027)** |
| 518 | Other diseases of lung | 85222.22 | 0.629 | 0.980 (0.975, 0.985) | | | |
| 584 | Acute kidney failure | 85241.07 | 0.634 | 0.983 (0.978, 0.987) | 95463.24 | 0.617 | 0.983 (0.978, 0.987) |
| **840** | **Sprains and strains of shoulder and upper arm** | | | | **95428.93** | **0.593** | **1.026 (1.022, 1.029)** |
| **845** | **Sprains and strains of ankle and foot** | | | | **95428.29** | **0.580** | **1.024 (1.021, 1.028)** |
| **847** | **Sprains and strains of other and unspecified parts of back** | **85236.47** | **0.588** | **1.066 (1.060, 1.071)** | **95405.80** | **0.597** | **1.066 (1.060, 1.071)** |
| **E920** | **Accidents caused by cutting and piercing instruments or objects** | | | | **95429.22** | **0.604** | **1.028 (1.025, 1.032)** |
| **E927** | **Overexertion and strenuous movements** | **85176.63** | **0.617** | **1.068 (1.063, 1.072)** | **95320.26** | **0.625** | **1.068 (1.063, 1.072)** |

**Table 4. Comparison of model fits (by 10-fold cross-validation) to predict the injury recurrence (at any time point after, ignoring time-to-event and censoring).**

| Model | AUC (95% CI) | sensitivity | specificity | cutoff |
|---|---|---|---|---|
| Logistic Regression with LASSO | 0.606 (0.597, 0.615) | 0.57 | 0.57 | 0.05 |
| Random Forest | 0.576 (0.566, 0.585) | 0.55 | 0.56 | 0.05 |
| ADTree | 0.600 (0.591, 0.606) | 0.55 | 0.58 | 0.18 |

displays in details the ranking values and confidence intervals for the top variables found in each comparison group that were selected by at least two ranking methods, highlighting those that were more frequently associated to the WPI.

When analyzing injury recurrence, the Kaplan-Meier estimate yielded the injury recurrence probability of 66.6% at the second year, 44.4% at the third year, and 13.4% at the fourth year for the WPI group.

The predictive models for injury occurrence exhibited moderate performance. The 'ever recurrence' models yielded a cross-validated AUROC between 0.57 and 0.60 (**Table 4**), while the survival models yielded a cross-validated c-index between 0.55 and 0.62 (**Table 5**). Overall, the Cox regression with stepwise selection exhibited the highest AUROC, and the variables of this model along with their hazard ratios (HRs) are listed in **Table 6**.

## Discussion

In this work, we investigated the health consequences associated with WPI and explored factors that may predict future recurrence of WPI in a large longitudinal statewide data set.

We found that WPI group showed higher proportions of people from Black African American and Hispanic ancestry, male, younger, who lived in areas with a higher deprivation index. The WPI group also included higher proportions of self-payers other than federal or private insurance. These findings were consistent with previous reports from national surveys in the U.S. [12,13].

The WPI group had the worse survival probability, after adjusting for age, compared to IOP and random controls; this confirms prior findings [14,15]. In this population, the three-year survival rate in WPI is 85.8%. We reckon that patients in IOP and WPI have different age distribution because of the employment ages. Although we had not matched ages in the design, we included only adult individuals; yet, there might be still a difference due to younger–not yet employed–adults and older–retired–adults.

In addition to higher mortality, people in the WPI group were also associated with higher risk of physical health morbidity. We used a robust framework to determine the importance of clinical consequences by combining three distinct measurements obtained from regression models. Compared against both IOP and random control groups, patients suffering WPI were more likely to be admitted into care, after injury, for acute bronchitis and bronchiolitis (ICD-

**Table 5. Comparison of model fits (by 10-fold cross-validation) to predict the injury recurrence using survival time-to-event and censoring set-up.**

| Model | Year | AUC | c-index (SE) |
|---|---|---|---|
| Survival tree | 1 | 0.542 | 0.55 (0.002) |
| | 2 | 0.546 | |
| | 3 | 0.552 | |
| Cox regression (stepwise selection) | 1 | 0.599 | 0.62 (0.005) |
| | 2 | 0.600 | |
| | 3 | 0.605 | |

**Table 6. Predictors of WPI recurrence as selected by stepwise Cox regression.**

| variable | HR (95% CI) | p-value |
|---|---|---|
| age | 0.99 (0.98, 0.99) | <0.0001 |
| insurance | 1.15 (1.12, 1.17) | <0.0001 |
| female | 0.75 (0.70, 0.80) | <0.0001 |
| (E927) Overexertion and strenuous movements | 1.28 (1.18, 1.39) | <0.0001 |
| (959) Injury other and unspecified | 1.26 (1.14, 1.40) | <0.0001 |
| (E917) Striking against or struck accidentally by objects or persons | 1.20 (1.10, 1.32) | <0.0001 |
| (E920) Accidents caused by cutting and piercing instruments or objects | 1.19 (1.09, 1.31) | 0.0002 |
| (36.07) Insertion of Drug-Eluting Coronary Artery Stent(s) | 0.35 (0.20, 0.61) | 0.0002 |
| (729) Other disorders of soft tissues | 1.14 (1.05, 1.24) | 0.0020 |
| (E000) External cause status | 1.31 (1.13, 1.52) | 0.0004 |
| (558) Other and unspecified noninfectious gastroenteritis and colitis | 1.21 (1.08, 1.35) | 0.0007 |
| (416) Chronic pulmonary heart disease | 0.33 (0.15, 0.74) | 0.0068 |
| (786) Symptoms involving respiratory system and other chest symptoms | 1.13 (1.05, 1.21) | 0.0009 |
| (99.38) Administration of Tetanus Toxoid | 1.29 (1.10, 1.52) | 0.0023 |
| (654) Abnormality of organs and soft tissues of pelvis | 0.65 (0.50, 0.85) | 0.0016 |
| (E916) Struck accidentally by falling object | 1.27 (1.09, 1.49) | 0.0011 |
| (81.92) Injection of Therapeutic Substance into Joint, or Ligament | 1.68 (1.23, 2.30) | 0.0027 |
| (E906) Other injury caused by animals | 1.23 (1.08, 1.42) | 0.0075 |
| (593) Other disorders of kidney and ureter | 0.71 (0.55, 0.91) | 0.0024 |

9: 466), sprains and strains of other and unspecified parts of back (ICD-9: 847), and overexertion and strenuous movements (ICD-9: E927). In addition to these conditions, when compared with random controls, the WPI group had higher odds of having sprains and strains at other body parts such as shoulder and upper arm (ICD-9: 840), and ankle and foot (ICD-9: 847). Our observations are consistent with prior findings that show how WPI lead to physical injuries [16]. Also, studies have reported that occupational exposure to various substances such as silica dust, gas, and fumes is related with the occurrence of chronic obstructive pulmonary disease (COPD) and related illnesses in the spectrum, as chronic bronchitis [17–19]. It is notable that the known associations are with chronic illness rather than acute, which is instead what we found. There is a number of possible explanations to this: 1) as we analyzed the first WPI and a limited, censored follow up time, a diagnosis of chronic bronchitis might not have been made yet, but recurring attacks of acute bronchitis may lead to chronic bronchitis; 2) our study population included both routine care and acute care, i.e. emergency rooms and urgent care centers, where an attack of bronchitis could be diagnosed as acute even in presence of an underlying condition; 3) possible selection bias, i.e. people with chronic bronchitis would have increased risk to be in care regardless the injury type, but acute episodes are differently distributed.

It is recognized that re-occurrence of incidents with a similar cause and circumstance in the workplace environment is a public health concern with unacceptable high incidence in the U. S. and worldwide [20]. Both pre-injury and post-injury correlates include social (disparity) determinants as race, and clinical conditions such as mental health disorders and drug dependence (which can be ascertained by prior ICD diagnoses) [21,22]. The Cox regression highlighted a number of conditions that affect the risk of WPI recurrence (**Table 6**). We identified a number of factors in the sociodemographic and clinical domains (e.g. age, insurance, gender, extant physical injury, chronic pulmonary conditions), but the prediction models did not yield good prediction performance. One of the reasons is that prior clinical history and

basic sociodemographics may not be the most informative domains to predict risk of WPI. Other predictors explaining a larger portion of variance could include job type, workplace safety, specific post-WPI work conditions, et cetera, which are not present in the HCUP data base.

This study has some limitations. First, heterogeneity in WPI was not accounted for in our study, as we used a group of ICD-9 codes to define the WPI group, which included all "accidents occurring in industrial places and premises", mine and quarry accidents, farm accidents, and accidents involving powered vehicles used solely within the buildings and premises of industrial or commercial establishment. These code do not differentiate the types of these industrial places and premises. Different types of industrial places and premises may have different impacts and risks for the future injury occurrence, but such information was not available in the data we used. In addition, mine and quarry accidents, and farm accidents were considered as WPI, while accidents occurring in unspecified place were considered as IOP. These are not 'explicitly' WPI, so it could introduce potential selection bias to our study; of note, the frequency of such codes was very low. Second, we used only ICD-9 codes for clinical diagnosis and procedures. Although the results of using such taxonomy can be directly applied to other electronic health record systems or similar data bases, if we want to fully understand the mechanics of how the predictors work and how to intervene on the identified predictors, clinical interpretation of these codes is still needed. In addition, in our analysis, short-term consequences and long-term consequences were combined together, i.e. we did not differentiate whether a diagnosis was made shortly after the injury or years after the injury.

Despite these limitations, we conducted a comprehensive analysis on the health consequences of and survival from workplace injuries, and their recurrence. Since the people's demographic and clinical features are responsible for a small portion of total recurrence risk, we reiterate the recommendation of the National Institute for Occupational Safety and Health to examine multiple information domains, especially the social and the ecological determinants, given the important role we found of the racial, health insurance and area deprivation distributions.

## Author Contributions

**Conceptualization:** Mattia Prosperi, Jiang Bian.

**Data curation:** Zhaoyi Chen.

**Formal analysis:** Zhaoyi Chen.

**Methodology:** Mattia Prosperi, Jiang Bian.

**Supervision:** Mattia Prosperi.

**Writing – original draft:** Zhaoyi Chen, Mattia Prosperi, Jiang Bian, Mo Wang, Chang Li.

**Writing – review & editing:** Zhaoyi Chen, Mattia Prosperi, Jiang Bian, Jae Min, Mo Wang, Chang Li.

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
