## [Decision Letter · Decision Letter 0]

13 Aug 2019

PONE-D-19-21474

Clinical correlates of workplace injury occurrence and recurrence in adults

PLOS ONE

Dear Chen,

Thank you for submitting your manuscript to PLOS ONE. After careful consideration, we feel that it has merit but does not fully meet PLOS ONE’s publication criteria as it currently stands. Therefore, we invite you to submit a revised version of the manuscript that addresses the points raised during the review process.

Both reviewers had several concerns, especially regarding the definition of the study groups and statistical analysis. I hope that the authors can effectively respond to their comments in the revised manuscript.

We would appreciate receiving your revised manuscript by Sep 27 2019 11:59PM. To enhance the reproducibility of your results, we recommend that if applicable you deposit your laboratory protocols in protocols.io, where a protocol can be assigned its own identifier (DOI) such that it can be cited independently in the future. For instructions see: http://journals.plos.org/plosone/s/submission-guidelines#loc-laboratory-protocols

We look forward to receiving your revised manuscript.

Kind regards,

Yu Ru Kou, PhD

Academic Editor

PLOS ONE

Journal Requirements:

3. Please provide a reworded financial disclosure which states specifically whether the funders played any role in the study.

If any authors received a salary from any of your funders, please state which authors and which funder. If the funders had no role, please state: "The funders had no role in study design, data collection and analysis, decision to publish, or preparation of the manuscript."

4. Please carefully proofread your manuscript for typographical errors. For example, in the abstract “We included adult who…” should be “We included adults who…”.

Reviewers' comments:

Reviewer's Responses to Questions

**Comments to the Author**

1. Is the manuscript technically sound, and do the data support the conclusions?

Reviewer #1: Yes

Reviewer #2: Partly

2. Has the statistical analysis been performed appropriately and rigorously? 

Reviewer #1: Yes

Reviewer #2: Yes

3. Have the authors made all data underlying the findings in their manuscript fully available?

Reviewer #1: Yes

Reviewer #2: Yes

4. Is the manuscript presented in an intelligible fashion and written in standard English?

Reviewer #1: Yes

Reviewer #2: Yes

5. Review Comments to the Author

Reviewer #1: Major revision

1.If one patient had a workplace injury, injury at other places, and without injury information in database. Which group did the patient belong to?

2.If one patient had more than two workplace injuries, which injury was selected in this study?

3.This study only matched on the distribution of diagnosis year, and why basic demographic data (gender, age, and race) not included in the matching criteria?

Reviewer #2: The study by Chen et al. is a case-control study combining three different databases and discussing the morbidities associated with workplace injury. They compared the characteristics of three groups include workplace injury, injury at other places and controls; concluding that WPI can lead to serious physical comorbidity burdens. They also built prediction models for injury recurrences, identifying several risk factors.

I have some comments as below:

1. In this study, workplace injury (WPI) was defined by ICD-9 diagnostic code E849.3, while diagnoses codes E849.0 to E849.9 except E849.3 were defined as injuries at other place (IOP). In general practice of occupational medicine, E849.2 (Mine and quarry accidents), E849.1 (Farm accidents) and even E849.9 (Accidents occurring in unspecified place) could also be used to describe WPI. Based on current definition, I think some WPI were actually included in IOP. This might affect your following models.

2. Since you have used ICD-9 code to define WPI group, how did you define the status of WPI recurrence. Please describe the detail in methods.

3. I suggest a better work to explain the relative low model performance and the variables that negatively associated with WPI (such as chronic pulmonary heart disease HR 0.33).

4. In the current study, you found that WPI were more likely to be admitted for acute bronchitis and bronchiolitis (ICD-9: 466). And you suggested that this could be related to COPD of occupational exposure. However, to our knowledge, occupational exposure related COPD was mostly in presence of chronic bronchitis instead of the ICD-9:466 you indicated.

6. PLOS authors have the option to publish the peer review history of their article (what does this mean?). If published, this will include your full peer review and any attached files.

Reviewer #1: No

Reviewer #2: No

---

## [Author Response · Author response to Decision Letter 0]

22 Aug 2019

Dear editor, thanks for considering our manuscript; we appreciated the reviewers’ feedback. We updated our manuscript in accordance to their suggestions, and prepared a line-by-line response letter showing the revisions made.

1. If one patient had a workplace injury, injury at other places, and without injury information in database. Which group did the patient belong to?

Re: As long as a patient had one WPI, it is assigned to the WPI group. If any IOP occurred before, it is accounted for as a possible predictor. We now have clarified the process in the Methods section.

2. If one patient had more than two workplace injuries, which injury was selected in this study?

It is the first recorded in the data base; the criterion is now stated in the Methods.

3. This study only matched on the distribution of diagnosis year, and why basic demographic data (gender, age, and race) not included in the matching criteria?

Re: Because it was of interest to describe --if any-- the demographic differences. In fact, we did find differences in these variables between WPI and the comparison groups. We reckon that IOP and WPI could be different among age groups because of the employment ages. Nonetheless, we included only adult individuals, so the putative bias might be only among younger (not yet employed) or older (retired) people. The Discussion sections now contains additional and more detailed considerations in this regard.

Reviewer #2: 

1. In this study, workplace injury (WPI) was defined by ICD-9 diagnostic code E849.3, while diagnoses codes E849.0 to E849.9 except E849.3 were defined as injuries at other place (IOP). In general practice of occupational medicine, E849.2 (Mine and quarry accidents), E849.1 (Farm accidents) and even E849.9 (Accidents occurring in unspecified place) could also be used to describe WPI. Based on current definition, I think some WPI were actually included in IOP. This might affect your following models.

Re: The reviewer was right and indeed we had included mine/quarry/farms accidents in the WPI group, and we clarified it in the methods. Thanks for pointing this out. We also discussed that an alternative categorization into IOP could be legit because they are not explicit. These codes are also not explicitly WPI –some of them might not be WPI– and this was the main reason we debated when we design the study on whether to keep them in or out; there would be misclassifications either way. We have checked the frequency of the corresponding ICD codes in the whole sample, and have reported them in the revised manuscript. The frequencies of these codes were rather low, thus, the potential misclassification of these code would have little impact on the overall study results (bearing the possibility of specific morbidities associated to these two injuries). We have updated the Discussion acknowledging this matter.

2. Since you have used ICD-9 code to define WPI group, how did you define the status of WPI recurrence. Please describe the detail in methods.

Re: WPI recurrence is defined as any WPI diagnosis recorded at least 30 days after the first WPI diagnosis. We decided to use a time window because a WPI diagnosis within the time window could have been a readmission for the same injury. The Centers for Medicare & Medicaid Services (CMS) define a hospital readmission as "an admission to an acute care hospital within 30 days of discharge from the same or another acute care hospital.”. We have added these details in Methods.

3. I suggest a better work to explain the relative low model performance and the variables that negatively associated with WPI (such as chronic pulmonary heart disease HR 0.33).

Re: The low model performance is likely due to the fact that prior clinical history and demographics may not be the most informative domains to predict risk of WPI, although extant theory incorporates mental health disorders and drug dependence as predictors (which can be ascertained by prior ICD diagnoses), and there is likely racial disparity. However, predictors that could explain a larger portion of variance include job type, workplace safety, physical stress, et cetera, which are not present in the HCUP data base. We have rewritten part of our Discussions section to better explain these other –unmeasured– contributors to overall risk.

4. In the current study, you found that WPI were more likely to be admitted for acute bronchitis and bronchiolitis (ICD-9: 466). And you suggested that this could be related to COPD of occupational exposure. However, to our knowledge, occupational exposure related COPD was mostly in presence of chronic bronchitis instead of the ICD-9:466 you indicated.

Re: the reviewer is right in reporting the association more likely with chronic bronchitis rather than acute. There is a number of possible explanations to this: 1) being this the first WPI, a diagnosis of chronic bronchitis might not have been made yet; 2) our population includes both routine care and acute care, i.e. emergency rooms and urgent care centers, where an attack of bronchitis would be diagnosed as acute even in presence of an underlying condition; 3) possible selection bias, i.e. people with chronic bronchitis have increased risk to be in care regardless the injury type, but acute episodes are differently distributed. We have added another paragraph in the Discussion accordingly.

---

## [Decision Letter · Decision Letter 1]

4 Sep 2019

[EXSCINDED]

Clinical correlates of workplace injury occurrence and recurrence in adults

PONE-D-19-21474R1

Dear Dr. Chen,

We are pleased to inform you that your manuscript has been judged scientifically suitable for publication and will be formally accepted for publication once it complies with all outstanding technical requirements.

With kind regards,

Yu Ru Kou, PhD

Academic Editor

PLOS ONE

Additional Editor Comments (optional):

Reviewers' comments:

Reviewer's Responses to Questions

**Comments to the Author**

1. If the authors have adequately addressed your comments raised in a previous round of review and you feel that this manuscript is now acceptable for publication, you may indicate that here to bypass the “Comments to the Author” section, enter your conflict of interest statement in the “Confidential to Editor” section, and submit your "Accept" recommendation.

Reviewer #1: All comments have been addressed

Reviewer #2: All comments have been addressed

2. Is the manuscript technically sound, and do the data support the conclusions?

Reviewer #1: Yes

Reviewer #2: Yes

3. Has the statistical analysis been performed appropriately and rigorously? 

Reviewer #1: Yes

Reviewer #2: Yes

4. Have the authors made all data underlying the findings in their manuscript fully available?

Reviewer #1: Yes

Reviewer #2: Yes

5. Is the manuscript presented in an intelligible fashion and written in standard English?

Reviewer #1: Yes

Reviewer #2: Yes

6. Review Comments to the Author

Reviewer #1: (No Response)

Reviewer #2: My major concern was the definition of the study groups. The authors had modified the definition and provided detailed results in the revision. All the comments have been addressed.

7. PLOS authors have the option to publish the peer review history of their article (what does this mean?). If published, this will include your full peer review and any attached files.

Reviewer #1: No

Reviewer #2: No

---

## [Editor Report · Acceptance letter]

6 Sep 2019

PONE-D-19-21474R1 

Clinical correlates of workplace injury occurrence and recurrence in adults 

Dear Dr. Chen:

I am pleased to inform you that your manuscript has been deemed suitable for publication in PLOS ONE. Congratulations! Your manuscript is now with our production department. 

With kind regards,

on behalf of

Dr. Yu Ru Kou 

Academic Editor

PLOS ONE